# Vaccination Against Herpes Zoster in Adults: Current Strategies in European Union Countries

**DOI:** 10.3390/vaccines13101073

**Published:** 2025-10-20

**Authors:** Manuela Chiavarini, Angela Bechini, Sara Boccalini, Alisa Barash, Enrica Castellana, Alessandro Senape, Paolo Bonanni

**Affiliations:** Department of Health Sciences, University of Florence, 50134 Florence, Italy; manuela.chiavarini@unifi.it (M.C.); angela.bechini@unifi.it (A.B.); enrica.castellana@unifi.it (E.C.); alessandro.senape@unifi.it (A.S.); paolo.bonanni@unifi.it (P.B.)

**Keywords:** immunization, prevention, *varicella zoster* virus, shingles, elderly, risk groups, recommendation

## Abstract

Background/Objectives: Herpes zoster (HZ), caused by varicella zoster virus (VZV) reactivation, significantly affects the functional status and quality of life of older adults and immunocompromised individuals. Vaccination represents an effective strategy to reduce the incidence of HZ. Methods: This review offers a cross-sectional assessment of the current landscape of adult herpes zoster vaccination strategies across the 27 EU member states, drawing on data available up to July 2025 from official sources such as the ECDC, the WHO, and national health authorities. Results: HZ vaccination is recommended in 17 EU countries (63%) according to the National Immunization Programs (NIPs) or by other institutional national health documents; in only 7 countries, vaccination is fully covered by the national healthcare system. HZ vaccination is recommended for healthy adults aged ≥50 years in 23.5% of countries (4/17), ≥60 years in 29.4% (5/17), and ≥65 years in 41.2% (7/17). At-risk groups are targeted in 94.1% of countries (16/17), predominantly from age 18 years (14 countries). Conclusions: An overall tendency toward broader HZ vaccination strategies, targeting both older adults and risk groups, is emerging. However, differences among national policies, together with the European Commission’s withdrawal of the live-attenuated Zostavax vaccine effective 1 June 2025, highlight the urgent need for comprehensive, harmonized immunization strategies to ensure adequate coverage of adult HZ vaccination across Europe.

## 1. Introduction

Herpes zoster (HZ), commonly called shingles, is caused by the reactivation of the varicella zoster virus (VZV), which, after primary infection (chickenpox/varicella), remains latent in the sensory ganglia. This reactivation can occur years later, as first described by Hope-Simpson in 1965 [1]. A decline in cell-mediated immunity is considered a key factor in VZV reactivation, providing a rationale for the increased incidence of HZ in immunocompromised individuals and the older subjects. Advanced age and immunosuppression—whether due to underlying conditions, immunosuppressive therapies, or mental health disorders such as depression—are well-established risk factors for herpes zoster [2,3]. These well-documented risk factors, along with their clinical relevance and management strategies, underscore the importance of recognizing the full clinical spectrum of herpes zoster, which includes not only the classic dermatomal vesicular eruption but also atypical manifestations such as systemic reactivation without cutaneous lesions (zoster sine herpete) [4,5].

HZ occurs worldwide, and the season does not influence the incidence rate. The incidence of HZ in Europe ranges from 2.0 to 4.6 per 1000 people per year in the general population [2]. The incidence of HZ varies with age, ranging from 1.2 to 3.4 cases per 1000 people per year in younger patients and increasing to 3.9–11.8 cases per 1000 individuals each year in the elderly population (i.e., over 65 years of age) [6].

HZ is a significant cause of disability among older adults and immunocompromised individuals, substantially affecting their quality of life as well as psychological and physical functioning aspects of patients’ lives [7]. Postherpetic neuralgia (PHN), defined as dermatomal neuropathic pain persisting for more than three months following the resolution of the acute herpetic rash, represents the most debilitating and therapeutically challenging complication of HZ, often lasting even years [8]. In addition to PHN, HZ is associated with a range of complications, including ophthalmic involvement, neurological sequelae, vascular and visceral complications, all contributing to increased hospitalization, mortality risk, healthcare costs and an economic burden in older adults [9]. Given its considerable burden, HZ remains a major public health concern. In this regard, particular emphasis should be placed on reducing disability and improving quality of life, especially in high-risk groups such as older adults and immunocompromised individuals. Broader implementation of HZ vaccination in these populations represents a key opportunity to mitigate this burden [7,10].

The availability of effective and well-tolerated HZ vaccines [11,12] represents a pivotal advancement in adult preventive medicine. Given the substantial clinical and socioeconomic burden associated with HZ, particularly due to its sharply increasing incidence in older adults, several European countries have included HZ vaccination into their national immunization recommendation to protect aging populations. In response to growing evidence on the effectiveness and safety of the recombinant HZ vaccine, the World Health Organization (WHO) has recently reviewed current data and issued updated recommendations on its use in public health programs [13]. The 2025 WHO recommendations for herpes zoster vaccination introduce several key updates. Notably, they recommend the adjuvanted recombinant vaccine (Shingrix) as the preferred option for preventing both herpes zoster and postherpetic neuralgia, even in immunocompromised individuals. According to the WHO, HZ vaccination is now advised for all adults aged ≥50 years, regardless of prior history of varicella zoster virus infection. Finally, the WHO emphasizes the importance of incorporating HZ vaccination into national immunization programs (NIPs), supported by updated evidence demonstrating superior efficacy and safety compared with the live-attenuated vaccine (Zostavax) [14]. These developments have prompted renewed attention to the status and structure of HZ immunization policies across Europe. Therefore, the objective of this study was to review and compare current national recommendations and reimbursement policies for adult HZ vaccination across EU member states, with particular attention to age-based and risk-based strategies.

## 2. Materials and Methods

A systematic search was conducted to collect and analyze data on HZ vaccination policies and strategies specifically across the 27 European Union (EU) member states. The study was designed as a cross-sectional review, providing a snapshot of the current status based on information available up to July 2025 from official sources.

The search was initiated using the European Centre for Disease Prevention and Control (ECDC) Vaccine Scheduler [15] as the primary source and was subsequently extended through targeted searches of institutional websites, including those of the WHO, national Ministries of Health, National Institutes of Health, and the peer-reviewed scientific literature. If information was not found in the above-mentioned institutional sources or there were uncertainties regarding the specific country, a literature review was conducted for the respective country. The search strategy employed the following terms: (“Herpes zooster”) AND ((immunization) OR (vaccination)). National immunization recommendations were further investigated through supplementary searches of official government and public health organization websites, employing structured queries via Google and institutional search engines.

Extracted data were compiled in a structured electronic database (Excel), which included the following variables for each country: current National Immunization Programs (NIP) recommendations, immunization schedules, age at vaccine administration, target risk groups, and funding mechanisms.

The population data for each country were extracted from the official statistics provided by the European Union, available at european-union.europa.eu (accessed on 2 October 2025).

To ensure data quality and consistency, each country profile underwent a two-stage validation process: initial double-checking by two independent researchers, followed by final review by a senior researcher. Two types of analyses were conducted: the evaluation of available vaccines and the review of immunization recommendations. The main characteristics of the available vaccines were examined, and collected data were subsequently analyzed to identify and compare key similarities and differences in vaccine availability and national immunization recommendations across EU member states.

## 3. Results

### 3.1. Herpes Zoster (HZ) Vaccines in Europe

Currently, two herpes zoster (HZ) vaccines are authorized for adult use in Europe by the European Medicines Agency (EMA).

Table 1 summarizes the key attributes of these vaccines, derived from the detailed technical characteristics provided by the European Medicines Agency (EMA) for the two authorized herpes zoster vaccines.

Zostavax is a live-attenuated vaccine (ZVL) first licensed in the United States of America (USA) in 2006 indicated for the prevention of herpes zoster and post-herpetic neuralgia in adults aged 50 years and older; it was administered as a single-dose schedule, but this vaccine is no longer in production and has now been replaced by an adjuvanted subunit HZ vaccine. Shingrix, a recombinant adjuvanted subunit vaccine (RZV) containing the varicella zoster virus glycoprotein E antigen, was developed and first licensed in Canada and the USA in 2017 and it is indicated for the prevention of herpes zoster and post-herpetic neuralgia in adults aged 50 years and older, as well as in adults 18 years and older at increased risk. Shingrix is given as a two-dose schedule, generally 2 to 6 months apart, and is preferred for its higher efficacy and longer-lasting protection. Several HZ vaccines have been developed, and candidates based on RNA and adenovirus-vectored platforms are in clinical development; however, the RZV is currently the only widely available vaccine [12].

### 3.2. HZ Vaccine Recommendations in EU Countries

This section provides an overview of the landscape of HZ vaccination recommendations for adults and vaccination schedules used in the EU European countries (Table 2).

HZ vaccination programs for adults have been recommended in 17 EU countries, including Austria, Belgium, Cyprus, Czechia, Denmark, Estonia, France, Germany, Greece, Ireland, Italy, Luxembourg, The Netherlands, Poland, Slovenia, Spain, and Sweden—representing 63.0% of EU member states (Figure 1). In 11 countries, it is included in the NIP (Austria, Belgium, Cyprus, Czechia, Estonia, France, Germany, Greece, Italy, Luxembourg, and Spain), and in 6 countries it is mentioned in specific documents (Denmark, Ireland, The Netherlands, Poland, Slovenia, and Sweden) (Table 2). Notably, in the remaining 10 countries (Bulgaria, Croatia, Finland, Hungary, Latvia, Lithuania, Malta, Portugal, Romania, and Slovakia), HZ vaccination is neither included in NIPs nor formally addressed within national immunization policies (Figure 1).

In only 7 countries (41.2%), Cyprus, France, Germany, Greece, Italy, Luxembourg and Spain, vaccination is offered free of charge by the national healthcare system (Table 2).

Among the 17 EU countries where HZ vaccination is recommended, national health authorities generally advise immunization for healthy adults aged ≥50 years in 23.5% of cases (4/17), ≥60 years in 29.4% (5/17), and ≥65 years in 35.3% (7/17) (Figure 2). In Denmark, although HZ vaccination is recommended, neither a specific target age group nor funding criteria based on age have been formally defined.

HZ vaccination is recommended for individuals classified as at risk in 94.1% of these countries (16/17), with 14 recommending immunizations from age 18 years (Austria, Belgium, Cyprus, Czechia, France, Greece, Ireland, Italy, Luxembourg, The Netherlands, Poland, Slovenia, Spain, and Sweden) and two from age 50 years (Denmark and Germany) (Figure 3).

Vaccination for at-risk groups is provided free of charge by the national healthcare system in 10 countries (58.5%): Austria, Belgium, Cyprus, Denmark, France, Germany, Greece, Italy, Luxembourg, and Spain. In 4 countries (23.5%)—Czechia, The Netherlands, Poland, and Slovenia—reimbursement for HZ vaccination is either partial or contingent upon insurance coverage. In Sweden, the decision to implement the recommendations and whether the vaccination cost is covered varies by region. In Ireland, the national healthcare system does not offer free HZ vaccination, resulting in out-of-pocket payments by individuals. In Estonia, although HZ vaccination is recommended, neither a specific at-risk group nor funding criteria based on risk group have been formally defined.

## 4. Discussion

As the global epidemiological landscape evolves, it is increasingly clear that adults—especially older individuals and those with chronic conditions—remain highly vulnerable to infectious diseases and their complications, underscoring the essential role of adult immunization in life-course vaccination strategies and modern public health policy [52,53,54,55]. A pertinent example of the importance of adult vaccination is the case of HZ, a reactivation of the varicella zoster virus that disproportionately affects older adults. The disease is associated with a significant burden of morbidity, including the risk of debilitating complications such as post-herpetic neuralgia. In this context, HZ vaccination represents a key preventive intervention, particularly for aging populations and individuals with underlying medical conditions [56]. Current immunization strategies increasingly focus on older adult populations through age-based approaches, highlighting the importance of a life-course vaccination framework aimed at reducing disease burden, preserving functional capacity, and improving health outcomes in later life. Global health authorities, including the WHO and the ECDC, emphasize the strategic expansion of adult immunization programs as a key component in mitigating the impact of vaccine-preventable diseases among aging population [57,58].

However, it is worth noting that 37% of EU countries do not include HZ vaccination in their NIPs or formally address it in national immunization policies, while the global benefits of vaccination are widely acknowledged, helping to reduce complications and alleviate the burden of healthcare assistance (such as hospitalizations, outpatient visits, and the need for medications) for both health services and society.

Over the past six years, HZ vaccination recommendations have grown by more than 140%, increasing from 7 countries in 2019 to 17 today—a clear public health success. Current recommendation of 63% is encouraging, though still below optimal levels [59]. Nonetheless, adult vaccination programs across Europe remain highly heterogeneous, both in age criteria and in the definition of “at-risk” individuals with underlying medical conditions. This persistent variability may hinder equitable access to HZ vaccination and challenge the achievement of broader adult immunization goals across Europe.

It is important to emphasize that the existence of recommendations does not necessarily imply that vaccination is being effectively administered [60].

Available literature points to significant variability in vaccination uptake, underscoring the critical need for collecting reliable coverage data to assess whether current strategies are effective or if additional interventions are required to promote vaccination uptake [61]. This highlights the necessity for robust surveillance systems and routine monitoring of vaccination coverage to better understand the real-world implementation of these recommendations and to guide public health actions aimed at improving vaccine access and acceptance. The systematic review by Sorrentino et al. provides a comprehensive examination of the logistic and organizational barriers to herpes zoster (HZ) vaccination in Europe, further underscoring the complexity of implementing vaccination programs [60]. The study identifies multiple critical challenges, including issues with healthcare professional engagement, gaps in patient awareness and perceptions of HZ burden, concerns regarding vaccine efficacy and safety, structural and accessibility limitations, and broader social dynamics that influence vaccine uptake. These findings emphasize the urgent need for targeted interventions that address both systemic and individual-level obstacles, such as educational campaigns for stakeholders, population groups, and healthcare providers, optimization of healthcare delivery structures, and strategies to improve patient outreach and acceptance. Together, these insights reinforce the importance of coordinated public health efforts, combining surveillance, targeted education, and structural improvements to enhance vaccination rates and mitigate the public health impact of HZ [60].

In contrast to the variability observed in adult immunization schedules across Europe, there is now increasing homogeneity regarding the type of HZ vaccine administered, largely due to regulatory developments. The European Commission has withdrawn the marketing authorization for the live-attenuated zoster vaccine, effective 1 June 2025, following the decision of the Company to permanently discontinue the marketing of the product for commercial reasons, marks a pivotal shift toward exclusive reliance on the recombinant zoster vaccine for HZ prevention across EU countries [12,13,14,15].

Specific adverse events after Zostavax that have led to its removal include several cases of disseminated vaccine virus infection, as reported by Li-Kim-Moy et al. [62] during the Australian nationwide vaccination campaign and by Kennedy and Grose [63], who identified ten serious adverse events in total associated with the live-attenuated zoster vaccine. These findings have reinforced the preference for the safer recombinant vaccine, Shingrix, which may prompt reconsideration of vaccination policies in some European countries.

This regulatory transition has substantial implications for national immunization strategies, necessitating revisions of clinical guidelines, adaptations in procurement policies, and targeted communication efforts to support continued uptake and ensure equitable access to RZV. Countries that previously offered both ZVL and RZV—such as Slovenia and Denmark—are now required to fully transition to RZV-based schedules, thereby aligning their HZ vaccination programs with evolving evidence and regulatory standards.

HZ revaccination may be indicated for individuals previously immunized with the live-attenuated zoster vaccine, particularly those at elevated risk for HZ and its complications. Several European countries have implemented RZV revaccination strategies for adults who previously received ZVL, reflecting the superior long-term efficacy of RZV compared with ZVL [15,29,49,50]. From a public health perspective, vaccination sessions should be carefully organized to accommodate the two-dose RZV schedule. In the United States, the Centers for Disease Control and Prevention (CDC), through the Advisory Committee on Immunization Practices (ACIP), recommends revaccination with RZV for adults aged ≥ 50 years, with a minimum interval of eight weeks between the prior ZVL dose and the first RZV dose [64]. This approach ensures optimal immunogenicity, maximizes protection against HZ and postherpetic neuralgia, and facilitates the implementation of structured revaccination programs.

Benefits include further reduction in HZ incidence, decreased risk of postherpetic neuralgia, and potential neurological and cardiovascular advantages. Italian data highlight the substantial economic burden of HZ in adults ≥ 50 years and in individuals with comorbidities, underscoring the value of preventive strategies [65]. From a public health perspective, revaccination programs require active recall, workforce training, and robust surveillance systems, particularly in European countries transitioning from ZVL, to maintain high levels of protection in aging populations [7].

However, the implementation of HZ vaccination strategies faces economic and logistical challenges, primarily due to the higher per-dose cost of the recombinant vaccine (Shingrix) compared with the previously available live-attenuated vaccine (Zostavax), as well as the additional resource requirements—including time, personnel, and organizational efforts—associated with Shingrix’s two-dose schedule versus the single-dose administration of Zostavax.

Nonetheless, recent evidence supports its economic value: the critical review by Giannelos et al. [66] found that most studies published between 2017 and 2022 (15 out of 18) identified RZV as cost-effective compared with no vaccination or ZVL, with some analyses even reporting cost-saving effects in immunocompromised populations. These considerations must be carefully integrated into national immunization programs and resource allocation frameworks, particularly given the increasing clinical vulnerability and healthcare demands of the aging population, while the current body of evidence on the cost-effectiveness of RZV can help inform decision-makers about the value of vaccination against herpes zoster [66].

This revision enhances the policy relevance and practical implications of our work by incorporating specific, evidence-based recommendations for EU health policymakers to facilitate more effective and targeted implementation of herpes zoster vaccination strategies. Recent real-world data further corroborate the effectiveness of the adjuvanted recombinant zoster vaccine (RZV) in populations at increased risk, reinforcing the rationale for these policy measures [67,68].

The high effectiveness of the adjuvanted recombinant zoster vaccine (RZV) was reported across both special and general populations. In a U.S. matched cohort study of adults ≥ 50 with autoimmune conditions (including rheumatoid arthritis, lupus, and multiple sclerosis), two doses of RZV reduced HZ incidence from approximately 12.9 to 4.3 per 1000 person-years, corresponding to 66.3% effectiveness (95% CI: 61.4–70.7), with variation by disease (e.g., ~48% for multiple sclerosis and ~77% for psoriasis) [67]. In the general population aged ≥ 50 years, a large U.S. prospective cohort study (~2 million individuals, 7.6 million person-years follow-up) reported RZV effectiveness of ~76% for two doses and ~64% for a single dose in preventing HZ [68].

This review has several limitations. We did not research data on vaccine coverage (VC): it was not the objective of the study, and these data are not available in most European countries. Therefore, a meaningful comparison between VC rates and national vaccination policies was not possible. The search process was influenced, in part, by the varying degrees of navigability and accessibility of official government websites across the countries surveyed. Additionally, language barriers and the limited accuracy of automated translation tools—such as Google Translate—posed challenges to the research team’s ability to thoroughly assess institutional content in its original language, constituting a significant methodological limitation.

In the context of the recent WHO position paper on herpes zoster vaccines, July 2025 [14], which provides updated global recommendations based on the latest evidence on the recombinant vaccine, and the scheduled market withdrawal of the live-attenuated Zostavax vaccine as of June 2025, this review offers timely and policy-relevant insights. By systematically analyzing national herpes zoster vaccination strategies across EU member states, the study contributes to a clearer understanding of the current landscape of adult immunization in Europe.

A key strength of this review is its comprehensive scope, which includes both age-based and risk-based recommendations, as well as the extent of vaccine reimbursement through national healthcare systems. This dual focus enables a nuanced assessment of how different countries prioritize and implement HZ vaccination, providing a valuable reference for policymakers seeking to align with evolving international guidance and to optimize protection for aging and vulnerable populations.

## 5. Conclusions

Herpes zoster vaccination offers benefits beyond individual protection, including improved quality of life and reduced healthcare costs. Strengthening adult immunization programs, particularly for high-risk groups such as older adults and individuals with comorbidities, is essential.

This review highlights that HZ vaccination is currently recommended in the majority of EU countries (17 out of 27), based on either NIPs or other national public health documents. However, full public reimbursement is provided in only 7 of these countries, potentially limiting equitable access to vaccination. While age-based recommendations vary, with thresholds ranging from ≥50 to ≥65 years, most countries (94.1%) also include at-risk populations starting from age 18 years, reflecting growing awareness of the disease burden in immunocompromised and chronically ill individuals.

Although a general trend toward broader HZ vaccination strategies is emerging, a certain variation persists across national policies. The recent withdrawal of the live-attenuated Zostavax vaccine by June 2025 further highlights the importance of updating immunization strategies and ensuring access to the most effective vaccines, such as the recombinant subunit vaccine, across the region.

These findings highlight the urgent need to advance coordinated, evidence-based adult immunization policies across Europe. Harmonizing recommendations and closing implementation gaps—particularly for age- and risk-based vaccination—must be recognized as public health priorities to reduce the burden of herpes zoster in aging and vulnerable populations. However, expanding recommendations alone is not sufficient: the next critical step is to ensure their effective adoption, monitoring, and surveillance within national health systems, thereby translating policy into measurable health gains.

## Figures and Tables

**Figure 1 vaccines-13-01073-f001:**
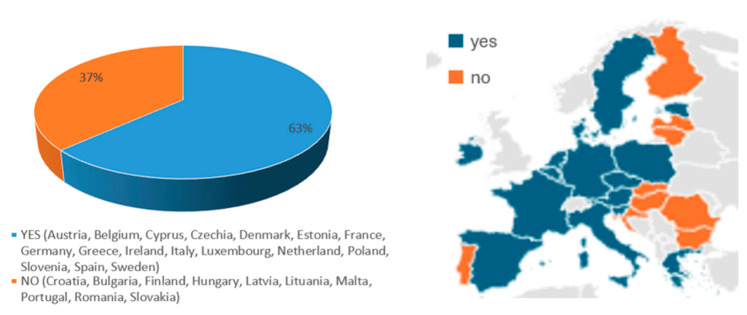
Existing HZ vaccination programs for adults in the 27 EU countries.

**Figure 2 vaccines-13-01073-f002:**
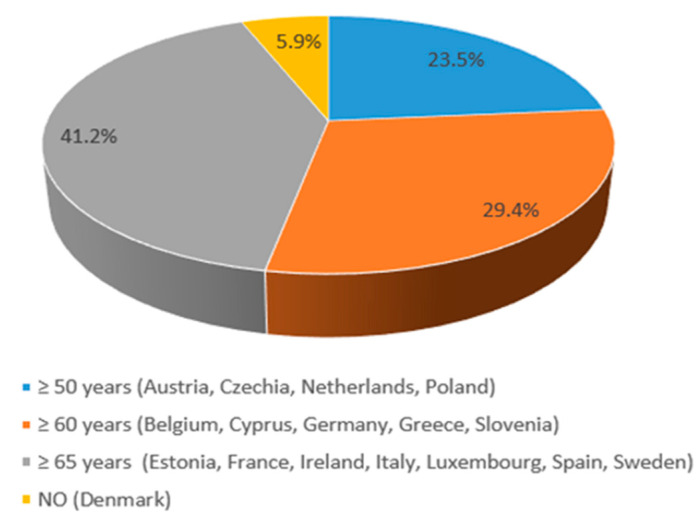
HZ vaccination programs for adults in EU countries by age.

**Figure 3 vaccines-13-01073-f003:**
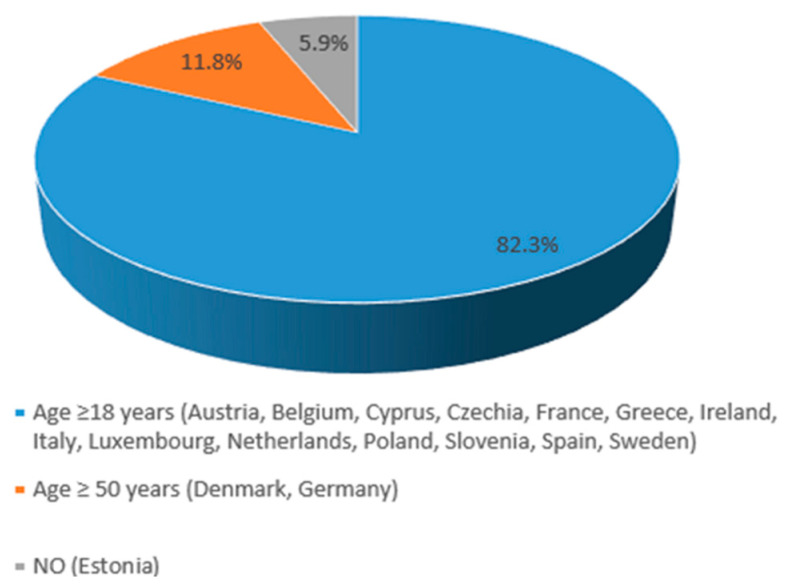
HZ vaccination programs for adults in high-risk groups in EU countries.

**Table 1 vaccines-13-01073-t001:** HZ vaccines authorized in the EU (Source: ECDC).

Vaccine Type (Acronym)	Description	Indication	Dose Schedule for Adults	Commercial Name (Company)	References
Live-attenuated vaccine (ZVL)	Contains a weakened form of varicella zoster virus (VZV)	Prevention of HZ and PHN in adults ≥ 50 years	Single dose	Zostavax (Merck/MSD- Boston, USA)	[12]
Recombinant subunit vaccine (RZV)	Non-live, contains recombinant glycoprotein E and AS01B adjuvant	Prevention of HZ and PHN in adults ≥ 50 years Prevention of HZ and PHN in adults ≥ 18 years at increased risk	Two doses (0 and 2–6 months apart)	Shingrix (GSK-London, UK)	[13]

**Table 2 vaccines-13-01073-t002:** HZ vaccine recommendations in EU countries.

Country Population (Inhabitants)	HZ Vaccination	Target population		Funding	References
		Eligible Age (years)	Minimum Age in Group(s) at Risk (Immunocompromised, Comorbidities)		
Austria (9,158,750)	YES (NIP)	50+	18+	NOT reimbursed by the national healthcare system, except for specific high-risk groups.	[15,16]
Belgium (11,763,650)	YES (NIP)	60+	18+	NOT reimbursed by the national healthcare system, except for specific high-risk groups.	[15,17]
Bulgaria (6,445,481)	NO				[15,18,19]
Croatia (3,861,967)	NO				[15,20,21]
Cyprus (966,400)	YES (NIP)	60+	18+	YES by national healthcare system	[15,22]
Czechia (10,900,555)	YES (NIP)	50+	18+	NOT reimbursed by the national healthcare system Depending on individual insurance providers or out-of-pocket payments for some adults.	[15,23,24]
Denmark (5,932,654)	YES Danish Health Authority		50+	Vaccine subsidies for certain groups of people (50+ age at high risk)	[15,25,26]
Estonia (1,369,995)	YES (NIP)	65+		NO: individuals typically bear the cost	[15]
Finland (5,400,000)	NO				[15,27]
France (68,600,000)	YES (NIP)	65+	18+	YES by the national healthcare system	[15,28]
Germany (84,700,000)	YES (NIP)	60+	50+	YES by the statutory health insurance (SHI) for the recommended target groups	[15,29]
Greece (10,400,000)	YES (NIP)	60+	18+	YES, by national healthcare system	[15,30]
Hungary (9,600,000)	NO				[15,31]
Ireland (5,380,000)	YES National Immunization Advisory Committee	65+	18+	NO	[15,32,33]
Italy (59,300,000)	YES (NIP)	65+	18+	YES, by the national healthcare system	[15,34,35]
Latvia (1,800,000)	NO				[15,36]
Lithuania (2,700,000)	NO				[15,37]
Luxembourg (660,000)	YES (NIP)	65+	18+	YES, by the national healthcare system	[15,38]
Malta (514,000)	NO				[15,39]
The Netherlands (17,700,000)	YES National Institute for Public Health and the Environment (RIVM); Ministry of Health, Welfare, and Sport.	50+	18+	YES	[15,40,41,42]
Poland (38,300,000)	YES Ministry of Health	50+	18+	Out-of-pocket for the vaccine at pharmacies.	[15,43]
Portugal (10,000,000)	NO				[15,44,45]
Romania (19,000,000)	NO				[15,46]
Slovakia (5,400,000)	NO				[15,47]
Slovenia (2,100,000)	YES National Institute of Public Health	60+	18+	For the most vulnerable immunocompromised people, vaccination is covered by compulsory health insurance. For others (including those aged 60 and over), vaccination is self-funded.	[15,48]
Spain (48,619,695)	YES (NIP)	65+	18+	YES, by the national healthcare system	[15,49,50]
Sweden (10,551,707)	YES The Public Health Agency	65+	18+	The regions can choose whether or not to implement the recommendations, and if the individual has to pay for the vaccination or not.	[15,51]

Belgium: However, many private health insurance policies cover part of the cost for individuals over 50 or for those with underlying health conditions. In some regions, discussions are ongoing about broader reimbursement policies. The Netherlands: Fully reimbursed, exclusively for insured individuals aged 18+ with specific high risk. By age group: reimbursement request for the basic health insurance package. Poland: However, starting in 2024, there is a 50% refund available for the RZV for individuals aged 65 and over who belong to specific risk groups.

## Data Availability

Please add the corresponding content of this part.

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
