# Peer review of "Vaccination Against Herpes Zoster in Adults: Current Strategies in European Union Countries"

_vaccines, 2025, doi:10.3390/vaccines13101073_

Round 1
Reviewer 1 Report
Comments and Suggestions for Authors
This manuscript is a review of herpes zoster vaccination in the EU countries. The review is timely because the EU Commission has withdrawn authorization for the live HZ vaccine called Zostavax in June 2025. Thus the only remaining EU zoster vaccine is the recombinant vaccine called Shingrix. Some recommendations for improvement of the manuscript are listed below. Also, more references should be cited.
1.Introduction, line 32. The authors are commended for the selection of the renown HZ article by Hope-Simpson as the first reference.
2.Introduction, line 52. Add a sentence to state that the association between herpes zoster ophthalmicus and subsequent ischemic stroke is especially strong. See article by H. Lin et al, Neurology 74:792, 2010. PMID: 20200348.
3.Figure 1, line 143. Please provide a new figure. For clarity to readers who do not live in the EU, draw a map of the EU rather than a circle. Name each country and use a different color for the countries that provide vaccination as contrasted with the counties that do not. The authors could even use a third color for countries that provide a free vaccination. Also list the population of each country somewhere in the text or figure.
4.Results, line 169. Add a new section to Results called Costs of vaccination. All vaccinologists should have a basic knowledge about costs of vaccination, even if the costs are being paid by a federal government. After a brief search, one article was found from Germany by D. Curran et al, Long term efficacy data for the recombinant zoster vaccine, Human Vaccines Immunother., 17: 5296, 2021. PMID: 34905463. The article quotes a German price of 110 euros per dose or 220 euros for a series of 2 doses of Shingrix. The current Italian authors are urged to check if they can find the costs in some of the other EU countries. At the very least, they should be able to tell us the costs to the Italian health department to provide Shingrix to Italian citizens. NOTE: In the Discussion (lines 241-52), the authors mention the important economic issues with vaccines, even though they never tell us the real costs of vaccines in the Results section,
5.Discussion about adverse events is too limited, line 215. The authors never mention specific adverse events after Zostavax that have led to its removal. The first report is written by J. Li-Kim-Moy et al, BMJ OPEN 13; 2023. PMID: 36707120. in a nationwide vaccination campaign with Zostavax in Australia, 3 Australian citizens died from a disseminated vaccine virus infection. In a second related article by P. Kennedy et al, Journal of Virology, 99:2025 (PMID:39818965), the authors surveyed the literature and found10 serious adverse events following Zostavax vaccination that have been published. The adverse events include 6 deaths, plus 4 nonfatal cases of disseminated vaccine virus infection. In addition to the 3 deaths in Australia, there has been 1 death each from the US, the UK and Canada. Perhaps some of the EU countries that do not offer shingles vaccination will change their recommendation when they read that Shingrix is a much safer vaccine to administer than Zostavax.
6.Conclusions. Line 275. Has any country in the EU published a paper to date that showed less herpes zoster in their population after immunization of their citizens with either Zostavax or Shingrix? Provide an answer in the manuscript and cite any manuscripts.
Author Response
Thank you for your copmments. The manuscript has been revised according to the reviewers’ feedback, leading to changes in the references, tables, figures, and line numbering
This manuscript is a review of herpes zoster vaccination in the EU countries. The review is timely because the EU Commission has withdrawn authorization for the live HZ vaccine called Zostavax in June 2025. Thus the only remaining EU zoster vaccine is the recombinant vaccine called Shingrix. Some recommendations for improvement of the manuscript are listed below. Also, more references should be cited.
We sincerely thank the reviewer for the positive and constructive evaluation of our manuscript. We greatly appreciate the acknowledgment of the relevance and contribution of our study. The responses to the specific comments are provided below.
1.Introduction, line 32. The authors are commended for the selection of the renown HZ article by Hope-Simpson as the first reference.
RESPONSE: Thank you very much for your kind comment.
2.Introduction, line 52. Add a sentence to state that the association between herpes zoster ophthalmicus and subsequent ischemic stroke is especially strong. See article by H. Lin et al, Neurology 74:792, 2010. PMID: 20200348.
RESPONSE: Thank you for the suggestion. The concept of the particularly strong association between herpes zoster ophthalmicus and subsequent ischemic stroke has been incorporated into the Introduction, along with the relevant bibliographic reference: "In addition to postherpetic neuralgia, herpes zoster is associated with a range of complications—including ophthalmic involvement, neurological sequelae, and vascular and visceral manifestations—with herpes zoster ophthalmicus being particularly strongly linked to subsequent ischemic stroke (8,9). These complications collectively contribute to increased hospitalization, higher mortality risk, elevated healthcare costs, and a substantial economic burden in older adults." (10. Lin HC, Chien CW, Ho JD. Herpes zoster ophthalmicus and the risk of stroke: a population-based follow-up study. Neurology. 2010 Mar 9;74(10):792-7. doi: 10.1212/WNL.0b013e3181d31e5c. Epub 2010 Mar 3. PMID: 20200348)
3.Figure 1, line 143. Please provide a new figure. For clarity to readers who do not live in the EU, draw a map of the EU rather than a circle. Name each country and use a different color for the countries that provide vaccination as contrasted with the counties that do not. The authors could even use a third color for countries that provide a free vaccination. Also list the population of each country somewhere in the text or figure.
RESPONSE: We thank the reviewer for this very helpful suggestion. To enhance clarity for readers who may be less familiar with the EU context, we have now added the population size of each EU Member State (see Table 2). The population data for each country were extracted from the official statistics provided by the European Union, available at european-union.europa.eu.(Material and Methods). While we initially considered that an EU map alone might not fully illustrate the overall distribution of adult HZ vaccination programs across the 27 countries, we agree that such a visual would greatly improve geographic comprehension. To address both aspects, we have revised Figure 1 to include both a pie chart showing the overall proportions and an EU map highlighting each country—using different colors to indicate whether vaccination is provided or not provided.
4.Results, line 169. Add a new section to Results called Costs of vaccination. All vaccinologists should have a basic knowledge about costs of vaccination, even if the costs are being paid by a federal government. After a brief search, one article was found from Germany by D. Curran et al, Long term efficacy data for the recombinant zoster vaccine, Human Vaccines Immunother., 17: 5296, 2021. PMID: 34905463. The article quotes a German price of 110 euros per dose or 220 euros for a series of 2 doses of Shingrix. The current Italian authors are urged to check if they can find the costs in some of the other EU countries. At the very least, they should be able to tell us the costs to the Italian health department to provide Shingrix to Italian citizens. NOTE: In the Discussion (lines 241-52), the authors mention the important economic issues with vaccines, even though they never tell us the real costs of vaccines in the Results section,
RESPONSE: We thank the reviewer for this observation. As correctly noted, the Discussion section addresses the relevant economic considerations associated with HZ vaccination. However, specific cost data were not included, as the study focuses on policy and strategy aspects rather than on economic evaluation. The text refers to the generally higher cost of the recombinant vaccine (Shingrix) compared with the previously available live-attenuated vaccine (Zostavax), as well as to the additional resource requirements — in terms of time, personnel, and logistics — related to the two-dose schedule of Shingrix versus the single-dose administration of Zostavax. We deemed it unnecessary to introduce a new section in the Results entitled “Costs of vaccination”; instead, we elaborated on this aspect in the Discussion, in line with the reviewers’ recommendations.
5.Discussion about adverse events is too limited, line 215. The authors never mention specific adverse events after Zostavax that have led to its removal. The first report is written by J. Li-Kim-Moy et al, BMJ OPEN 13; 2023. PMID: 36707120 in a nationwide vaccination campaign with Zostavax in Australia, 3 Australian citizens died from a disseminated vaccine virus infection. In a second related article by P. Kennedy et al, Journal of Virology, 99:2025 (PMID:39818965), the authors surveyed the literature and found10 serious adverse events following Zostavax vaccination that have been published. The adverse events include 6 deaths, plus 4 nonfatal cases of disseminated vaccine virus infection. In addition to the 3 deaths in Australia, there has been 1 death each from the US, the UK and Canada. Perhaps some of the EU countries that do not offer shingles vaccination will change their recommendation when they read that Shingrix is a much safer vaccine to administer than Zostavax.
RESPONSE: We thank the reviewer for this observation. As suggested we insert “ Specific adverse events after Zostavax that have led to its removal include several cases of disseminated vaccine virus infection, as reported by Li-Kim-Moy et al.(Li-Kim-Moy J, Phillips A, Morgan A, Glover C, Jayasinghe S, Hull BP, Dey A, Beard FH, Hickie M, Macartney K. Disseminated varicella zoster virus infection following live attenuated herpes zoster vaccine: descriptive analysis of reports to Australia's spontaneous vaccine pharmacovigilance system, 2016-2020. BMJ Open. 2023 Jan 27;13(1):e067287. doi: 10.1136/bmjopen-2022-067287. PMID: 36707120; PMCID: PMC9884885) during the Australian nationwide vaccination campaign and by Kennedy and Grose (Kennedy PGE, Grose C. Insights into pathologic mechanisms occurring during serious adverse events following live zoster vaccination. J Virol. 2025 Feb 25;99(2):e0181624. doi: 10.1128/jvi.01816-24. Epub 2025 Jan 17. PMID: 39818965; PMCID: PMC1185280), who identified ten serious adverse events in total associated with the live attenuated zoster vaccine. These findings have reinforced the preference for the safer recombinant vaccine, Shingrix, which may prompt reconsideration of vaccination policies in some European countries”. Additionally, we have included a paragraph with specific, evidence-based recommendations for EU health policymakers derived from the study findings. (Constenla D, Lonnet G, Aris E, Ramsanjay RK, Servotte N, Mwakingwe-Omari A, Alsdurf H, Yun H. Real-world effectiveness of the adjuvanted recombinant zoster vaccine in ≥50-year-old adults with autoimmune diseases. J Infect Dis. 2025 Aug 11:jiaf395. doi: 10.1093/infdis/jiaf395. Epub ahead of print. PMID: 40795879. Zerbo O, Bartlett J, Fireman B, Lewis N, Goddard K, Dooling K, Duffy J, Glanz J, Naleway A, Donahue JG, Klein NP. Effectiveness of Recombinant Zoster Vaccine Against Herpes Zoster in a Real-World Setting. Ann Intern Med. 2024 Feb;177(2):189-195. doi: 10.7326/M23-2023. Epub 2024 Jan 9. PMID: 38190712; PMCID: PMC11001419). These revisions aim to strengthen the policy relevance and practical implications of our work. The revisions are highlighted in the updated manuscript for clarity
6.Conclusions. Line 275. Has any country in the EU published a paper to date that showed less herpes zoster in their population after immunization of their citizens with either Zostavax or Shingrix? Provide an answer in the manuscript and cite any manuscripts.
Response: Response: We thank the reviewer for this suggestion. We have addressed this point by adding relevant references and incorporating specific, evidence-based considerations for EU health policymakers in the Discussion section.
The manuscript has been revised according to the reviewers’ feedback, leading to changes in the references, tables, figures, and line numbering

Reviewer 2 Report
Comments and Suggestions for Authors
The clinical features of herpes zoster should be at least cited in the introduction of the article. The authors described the possible complication of herpes zoster but they neglected the description of the classic and less classic presentation of the disease (for example the possibility of the systemic reactivation of Varicella zoster virus without skin lesions - zoster sine herpete).
See for example:
Drago F, Herzum A, Ciccarese G, Broccolo F, Rebora A, Parodi A. Acute pain and postherpetic neuralgia related to Varicella zoster virus reactivation: Comparison between typical herpes zoster and zoster sine herpete. J Med Virol. 2019 Feb;91(2):287-295. doi: 10.1002/jmv.25304. Epub 2018 Sep 24. PMID: 30179265.
Examples of immunocompromised individuals who may benefit from anti-HZ vaccination (patients with autoimmune diseases, patients with human immunodeficiency virus infection, transplanted patients...) should be given in the text.
Author Response
Thank you for your comments. The manuscript has been revised according to the reviewers’ feedback, leading to changes in the references, tables, figures, and line numbering.
We sincerely thank the reviewer for the constructive evaluation of our manuscript and for suggestions, which allowed us to expand and strengthen the Introduction and Discussion section.
The clinical features of herpes zoster should be at least cited in the introduction of the article. The authors described the possible complication of herpes zoster but they neglected the description of the classic and less classic presentation of the disease (for example the possibility of the systemic reactivation of Varicella zoster virus without skin lesions - zoster sine herpete).
See for example:
Drago F, Herzum A, Ciccarese G, Broccolo F, Rebora A, Parodi A. Acute pain and postherpetic neuralgia related to Varicella zoster virus reactivation: Comparison between typical herpes zoster and zoster sine herpete. J Med Virol. 2019 Feb;91(2):287-295. doi: 10.1002/jmv.25304. Epub 2018 Sep 24. PMID: 30179265.
RESPONSE: Thank you for your comment. We have slightly expanded the section on the clinical features of herpes zoster. While the primary focus of our manuscript is Vaccination against Herpes Zoster in Adults: Current Strategies within European Union countries—thus emphasizing a public health rather than a purely clinical perspective—we have incorporated the following sentence in the introduction to acknowledge the broader clinical spectrum: "These well-documented risk factors, along with their clinical relevance and management strategies, underscore the importance of recognizing the full clinical spectrum of herpes zoster, which includes not only the classic dermatomal vesicular eruption but also atypical manifestations such as systemic reactivation without cutaneous lesions (zoster sine herpete)." (Patil A, Goldust M, Wollina U. Herpes zoster: A Review of Clinical Manifestations and Management. Viruses. 2022 Jan 19;14(2):192. doi: 10.3390/v14020192. PMID: 35215786; PMCID: PMC8876683; Drago F, Herzum A, Ciccarese G, Broccolo F, Rebora A, Parodi A. Acute pain and postherpetic neuralgia related to Varicella zoster virus reactivation: Comparison between typical herpes zoster and zoster sine herpete. J Med Virol. 2019 Feb;91(2):287-295. doi: 10.1002/jmv.25304. Epub 2018 Sep 24. PMID: 30179265).
Examples of immunocompromised individuals who may benefit from anti-HZ vaccination (patients with autoimmune diseases, patients with human immunodeficiency virus infection, transplanted patients...) should be given in the text.
Response: Recent real-world data have further reinforced the evidence supporting the effectiveness of the adjuvanted recombinant zoster vaccine (RZV) in populations at increased risk. In particular, Constenla DG et al. (2025) demonstrated that RZV provides substantial protection against herpes zoster in adults aged ≥50 years with autoimmune diseases, a group traditionally considered vulnerable due to immune dysregulation and frequent use of immunosuppressive therapies. The study showed a significant reduction in herpes zoster incidence and related complications, with a favorable safety profile consistent with previous clinical trials. Additionally, we have included a paragraph with specific, evidence-based recommendations for EU health policymakers derived from the study findings.
(Constenla DG, Lonnet G, Aris E, Ramsanjay RK, Servotte N, Mwakingwe-Omari A, Alsdurf H, Yun H. Real-world effectiveness of the adjuvanted recombinant zoster vaccine in ≥50-year-old adults with autoimmune diseases. J Infect Dis. 2025 Aug 11:jiaf395. doi: 10.1093/infdis/jiaf395. PMID: 40795879).
The manuscript has been revised according to the reviewers’ feedback, leading to changes in the references, tables, figures, and line numbering

Reviewer 3 Report
Comments and Suggestions for Authors
The paper by Chiavarini et al, clearly highlights the importance of adult vaccination against herpes zoster (HZ), especially among older adults and high-risk groups in European Union. Overall, the findings presented are valuable and contribute meaningfully to the existing knowledge. However, a few aspects of the manuscript could be further improved. I therefore suggest a minor revision to enhance the overall quality and clarity of the paper.
Comments are listed below:
Abstract:
- Lines 12–13: Clearly define the study design and specify the study period.
- Line 19: The statement about a “general trend” should be substantiated by showing the dynamics of the introduction of HZ vaccination recommendations in EU countries during the study period in the Results section.
Matherials and Methods:
- Define the study design and clearly indicate the study period (e.g., from month/year to July 2025).
- Suggestion: although the study focuses on recommendations within the EU, extending the analysis to the entire WHO European Region would have made the results more comprehensive. Since EU countries are generally more economically developed, including non-EU European countries would provide a broader and more balanced overview, helping to better identify regional differences across Europe.
Results:
Table 2:
- Revise the title of the fourth column to indicate the minimum age of vaccine recipients. In that column, remove additional explanations (e.g., Germany 50+ “at increased risk”).
• Simplify the content of some cells in the fifth column (Funding) — for example, for Belgium, Ireland, the Netherlands, and Poland — by moving additional notes to table footnotes.
• Avoid repeating results both in the text and in figures. This applies to all three figures. Either remove the numerical values from the text or keep only the figures.
Discussion:
- The discussion section (lines 171–192) should be more concise and avoid repeating general statements already presented in the Introduction.
- The authors should compare their main findings with those from previously published studies conducted in the EU.
- Lines 193–201: The manuscript notes challenges related to the implementation of recommendations for HZ vaccination; it would be useful to elaborate on the key factors influencing HZ vaccine implementation.
- Discuss the relationship between the inclusion of HZ vaccination in national immunization programs (NIPs), the availability of free vaccines, the quality of HZ surveillance, and the existence of cost–benefit analyses or burden-of-disease studies within the population.
Conclusion
- Consider adding specific, evidence-based recommendations for EU health policymakers derived from the study findings.
Author Response
Thank you for your comments. The manuscript has been revised according to the reviewers’ feedback, leading to changes in the references, tables, figures, and line numbering.
The paper by Chiavarini et al, clearly highlights the importance of adult vaccination against herpes zoster (HZ), especially among older adults and high-risk groups in European Union. Overall, the findings presented are valuable and contribute meaningfully to the existing knowledge. However, a few aspects of the manuscript could be further improved. I therefore suggest a minor revision to enhance the overall quality and clarity of the paper.
We sincerely thank the reviewer for the positive and constructive evaluation of our manuscript. We greatly appreciate the acknowledgment of the relevance and contribution of our study. The responses to the specific comments are provided below.
Abstract:
- Lines 12–13: Clearly define the study design and specify the study period.
- Line 19: The statement about a “general trend” should be substantiated by showing the dynamics of the introduction of HZ vaccination recommendations in EU countries during the study period in the Results section
Response: We thank the reviewer for this valuable comment. We fully agree that information on the dynamics of HZ vaccination policy development would be important to illustrate long-term trends. However, as specified in the Materials and Methods section, our study was designed as a cross-sectional review providing a snapshot of the current status of HZ vaccination strategies in EU countries, based on data collected from official sources (ECDC, WHO, and national health websites) up to July 2025. Therefore, temporal trends in the introduction of national recommendations were beyond the scope of our analysis. To clarify this point, we have revised the text accordingly.
Matherials and Methods:
- Define the study design and clearly indicate the study period (e.g., from month/year to July 2025).
- Suggestion: although the study focuses on recommendations within the EU, extending the analysis to the entire WHO European Region would have made the results more comprehensive. Since EU countries are generally more economically developed, including non-EU European countries would provide a broader and more balanced overview, helping to better identify regional differences across Europe.
Response: We thank the reviewer for this valuable comment. We fully agree that including non-EU countries within the WHO European Region would have provided a more comprehensive and balanced overview of HZ vaccination policies across Europe. However, as specified in the Materials and Methods section, our study was designed as a cross-sectional review focused specifically on the 27 EU member states, with the aim of providing a snapshot of the current status of HZ vaccination strategies based on data collected from official sources (ECDC, WHO, and national health websites) up to July 2025. Expanding the analysis to include non-EU European countries is certainly of great interest and could represent an important direction for future research.
Results:
Table 2:
- Revise the title of the fourth column to indicate the minimum age of vaccine recipients. In that column, remove additional explanations (e.g., Germany 50+ “at increased risk”).
• Simplify the content of some cells in the fifth column (Funding) — for example, for Belgium, Ireland, the Netherlands, and Poland — by moving additional notes to table footnotes.
• Avoid repeating results both in the text and in figures. This applies to all three figures. Either remove the numerical values from the text or keep only the figures.
Response: We thank the reviewer for these valuable suggestions. While we acknowledge that avoiding repetition between text and figures can improve conciseness, we believe that including key numerical values in the text enhances readability and helps guide the reader through the figures. All reported values directly refer to the corresponding figures, providing a clear and complementary presentation of the results. Table 2 has been revised to improve clarity and readability: the fourth column title now indicates the minimum age of vaccine recipients, with additional explanations removed; several cells in the fifth column (Funding) have been simplified, and extra notes have been moved to footnotes.
Discussion:
- The discussion section (lines 171–192) should be more concise and avoid repeating general statements already presented in the Introduction
- The authors should compare their main findings with those from previously published studies conducted in the EU.
- Lines 193–201: The manuscript notes challenges related to the implementation of recommendations for HZ vaccination; it would be useful to elaborate on the key factors influencing HZ vaccine implementation.
- Discuss the relationship between the inclusion of HZ vaccination in national immunization programs (NIPs), the availability of free vaccines, the quality of HZ surveillance, and the existence of cost–benefit analyses or burden-of-disease studies within the population.
Response: We thank the reviewer for the suggestion. As requested, we have made the first part of the Discussion more concise and have compared our main findings with previously published EU studies, as highlighted: “Over the past six years, HZ vaccination recommendations have grown by more than 140%, increasing from 7 countries in 2019 to 17 today—a clear public health success. Current recommendation coverage of 63% is encouraging, though still below optimal levels (Cassimos DC et al., 2020)”. We have also expanded the Discussion to address the implementation of HZ vaccination recommendations, elaborating on key factors such as inclusion in national immunization programs (NIPs), vaccine availability, surveillance quality, and supporting cost–benefit or burden-of-disease studies (Sorrentino M et al., 2024). Additionally, we have included a paragraph with specific, evidence-based recommendations for EU health policymakers derived from the study findings. (Constenla D et al., 2025; Zerbo O et al., 2024). These revisions aim to strengthen the policy relevance and practical implications of our work. The revisions are highlighted in the updated manuscript for clarity
Conclusion: Consider adding specific, evidence-based recommendations for EU health policymakers derived from the study findings.
Response: We thank the reviewer for this suggestion. We have addressed this point by adding relevant references and incorporating specific, evidence-based considerations for EU health policymakers in the Discussion section.
The manuscript has been revised according to the reviewers’ feedback, leading to changes in the references, tables, figures, and line numbering.
